# All-on-4 Hybrid with Extra-Long Transnasal Implants: Descriptions of the Technique and Short-Term Outcomes in Three Cases

**DOI:** 10.3390/jcm13113348

**Published:** 2024-06-06

**Authors:** Mariana Nunes, Miguel de Araújo Nobre, Vanderlim Camargo

**Affiliations:** 1Oral Surgery Department, Malo Clinic, Avenida dos Combatentes, 43, Level 9, 1600-042 Lisboa, Portugal; mnunes@maloclinics.com; 2R&D Department, Malo Clinic, Avenida dos Combatentes, 43, Level 11, 1600-042 Lisboa, Portugal; 3CIIPO, Rua Germano Wendhausen, 203-3 andar-Centro, Florianópolis 88015-460, Brazil; vanderlim@me.com

**Keywords:** transnasal, dental implants, zygomatic implants, all-on-4

## Abstract

**Background/Objectives**: There is a need for alternative approaches to full-arch rehabilitation of atrophic maxillae. The aim of this short case series was to describe the technique and assess the short term-outcomes of atrophic maxillae rehabilitation using transnasal implants in conjunction with zygomatic implants. **Methods**: Three female patients (average age: 62 years) presenting comorbidities and atrophic maxillae preventing the insertion of standard maxillary anchored implants received a full-arch fixed prosthesis supported by transnasal implants together with zygomatic implants, using the *ad modum* all-on-4 concept. Patients were followed during the functional osseointegration period. Primary outcome measures were prosthetic and implant survival based on function. Secondary outcome measures were complication parameters (biological and mechanical), plaque and bleeding levels, and probing pocket depths > 4 mm. **Results**: No implant failures were registered, and all prostheses remained in function. The only complication was a fracture of a provisional crown that was resolved. Plaque and bleeding scores were mild during the follow-up period. **Conclusions**: The present manuscript describes the use of extra-long transnasal implants in combination with zygomatic implants in immediate function for full-arch fixed prosthetic rehabilitation of atrophic maxillae, with the objective of promoting more research into this relatively recent technique. More studies are needed to validate the technique.

## 1. Introduction

Edentulism is a major problem in oral health, causing biological, psychological, and social disorders. Globally, according to the World Health Organization, 30% of individuals aged between 65 and 74 years do not have any natural teeth in the oral cavity [1]. Alveolar bone resorption is a chronic, progressive, and cumulative multifactorial process of bone remodeling and is an inevitable consequence of tooth loss [2], resulting in atrophic jaws that are challenging to rehabilitate. The “all-on-4” concept (Nobel Biocare AB, Gothenburg, Sweden) is a treatment protocol for rehabilitating edentulous jaws through a fixed prosthesis supported by four implants in immediate function, with two anterior implants axially positioned and two posterior implants distally tilted [3,4]. This approach has registered good results in both short- and long-term follow-up, with cumulative success rates ranging between 91.7% and 99.7% in both arches [3,4]. 

To apply the all-on-4 concept (Nobel Biocare AB) through a standard approach, a minimum bone availability is necessary, which, in situations of severe bone resorption and/or exacerbated maxillary sinus pneumatization, requires alternative approaches [5,6]: The use of long implants with trans-sinusal anchorage [6], short implants [7], or zygomatic implants [8,9,10,11,12] represent viable alternatives to resolve this limitation while maintaining the approach of the *ad modum* all-on-4 concept (Nobel Biocare AB). Moreover, these techniques could also be combined with the use of bone grafts [13,14]. 

Among the various treatment options, the use of zygomatic implants has proven to be an approach with high success rates over the last 30 years [15,16,17,18]. Depending on the degree of maxillary bone atrophy, the zygomatic implant may be used in combination with one to three conventional anterior implants (all-on-4 hybrid) [19], or in cases of inadequate bone volume preventing the insertion of conventional dental implants, using four zygomatic implants (quad zygoma; all-on-4 double zygoma) [20,21]. The all-on-4 double zygoma surgical procedure is a sensitive technique and requires advanced surgical skills. An alternative to the use of four zygomatic implants could involve placing two extra-long transnasal implants associated with unilateral zygomatic implants [22,23].

The aim of this case series was to describe the technique for the rehabilitation of atrophic maxillae with fixed prosthetic full-arch restorations supported by extra-long transnasal implants associated with unilateral zygomatic implants, illustrating the scope of possible indications, contra-indications, planning, and surgical protocol.

## 2. Materials and Methods

This retrospective short case series (n = 3 cases) illustrates the surgical protocol for using transnasal implants (Vanderlim technique) [24,25] in conjunction with zygomatic implants in full-arch fixed prosthetic rehabilitations with the *ad modum* all-on-4 concept. The patients were treated in a private practice (MALO CLINIC, Lisbon, Portugal), and the protocol was approved by an institutional clinical/human experimentation panel (Ethical Committee for Health, Lisbon, Portugal, authorization no. 001/2024). The treatment was carried out respecting the Helsinki Declaration of 1964, as revised in 2013, with the patients providing written informed consent to participate. 

### 2.1. Indications and Contra-Indications

The full-arch maxillary rehabilitation using transnasal implants in conjunction with zygomatic implants requires the following anatomical circumstances: impossibility of performing a “standard” all-on-4 due to the absence of bone available to insert four standard implants in the maxillary arch; a posterior maxillary region with a bone quantity of D-V or D-VI (Cawood–Howell classification) [26]; a minimum bone height of 4 mm between the maxillary ridge and the nasal cavity, and a minimum of 3 mm on the frontal process of the maxilla—“crista conchalis corporis maxilla” for the apical anchorage of extra-long implants [22,23]

Standing as an anatomical limitation is the presence of a very wide nasal cavity with distally placed lateral limits—increased nose antrum [22].

### 2.2. Planning

A clinical examination with a pre-operative orthopantomography and a cone beam computed tomography (CBCT) scan were used to plan the surgery. The radiological record data were exported in a DICOM3 (Digital Imaging and Communication On Medicine) format file to the DTX Studio Implant software (V.3.6.4.2, Nobel Biocare AB), which was used for digital planning. The axial axis was displaced until the zygomatic arch became completely visible and was marked using the software tools. During simulation, it was possible to confirm the crestal bone availability, the frontal process prominence (crista conchalis of the maxilla), and the zygomatic bone, enabling us to perform virtual implant placement of both the conventional and extra-long transnasal and zygomatic implants [27]. The position of the nasolacrimal duct was carefully evaluated given the location of its foramen (under the inferior nasal turbinate, relatively close to the conchal crest). In a situation of enough bone available on the pre-maxilla [19] to place a conventional implant (bone quantity of D-V or D-VI—Cawood–Howell classification, minimum of 7 mm) [26], that option was attempted first. The Nasal Anatomic Systematic Approach (NASA) was used to classify the patients according to the crestal approach for transnasal implant insertion (NASA type 0: crestal approach from the foramen; NASA type 1: crestal approach from the central incisor; type 2: crestal approach from the lateral incisor; type 3: crestal approach from the canine; type 4: crestal approach from the canine with a screw thread exposed) [24]. NobelZygoma implants (Nobel Biocare AB) were chosen from the software’s implant library and were used to perform the virtual surgical planning over the radiological data. Similarly, extra-long transnasal implants (22.5 mm of length) were selected from the library and their position was planned in the software. Once the virtual planning was performed, the stereolithographic of the middle third of the face with the maxilla extended was printed before the surgical procedure, for safer planning and manual training. To allow printing, the DICOM files were exported to the Blue Sky Plan software (V4.9.4 64 bit, BlueSkyBio, Libertyville, IL, USA), where the models were converted to STL file extensions by editing. The impressions were 3D printed (Formlabs Inc., Form 3B+, Somerville, MA, USA) using a photopolymer resin.

### 2.3. Surgical and Prosthetic Procedures 

The implant surgery was performed by the same surgeon (M.N.), with the first patient rehabilitated in January 2021 and the last patient in July 2022. 

All surgeries were performed under general anesthesia, with nasal intubation. Infiltrative anesthesia with a vasoconstrictor was administered to reduce the bleeding and post-operatorive pain.

A mucoperiosteal incision was made along the crest of the ridge, slightly palatal, from the molar to the contralateral molar area, with buccal vertical releasing incisions made in the midline, and posteriorly to expose the zygomatico-maxillary buttress and the prominence of the zygoma. Flap reflection allowed for infra-orbital nerve identification and protection as well as direct observation of the lateral aspect of the zygomatic bone. The anterior reflection of the flap exposed the nasal cavity and allowed the detachment of its lateral portion up to the height of the piriform aperture, exposing the nasal mucosa. Using a maxillary sinus lift detacher (Hu-Friedy, Leimen, Germany) and Lucas curettes (Hu-Friedy, Germany), the nasal cavity’s lateral wall and floor were exposed by detaching the nasal mucosa. The palatal mucosa was also reflected. 

The planning on the 3D models aided in determining the beginning of the osteotomy for the placement of the extra-long transnasal implants and was pointed directly at the bone using a bone marker. A surgical kit composed of extra-long burs (Helix Compact Surgical Kit GM Long, Neodent, Curitiba, Brazil) was used. Using a drilling speed of 850 RPM, the osteotomy was initiated on the crest, with the spear lightly touching the lateral wall of the nasal cavity according to the planning in the DTX Studio Implant software (V.3.6.4.2, Nobel Biocare AB, Sweden), and proceeded in the direction of the maxilla’s frontal process. The nasal membrane was maintained away from the drilling by using a periosteal (Modified Austin Tissue Retractor, Hu-Friedy, Germany), which assisted in the direct visualization of the internal path. Following the protocol, 2.5 and 3.75 mm drills were used. Despite the need for high primary stability to perform the immediate function, underpreparation was not performed due to the risk of fracture [28]. The implant choice was based on two characteristics considered essential for the use in this region: sufficient length (minimum 21 mm) and a narrow diameter, in order to be accommodated in the thin region of the maxilla’s frontal process. The exact length was only defined after performing the osteotomy and probing the alveolus. Helix GM^®^ Long implants (Neodent, Curitiba, Brazil) with 22.5 mm of length were placed, and in order to avoid nasal mucosal adherence to the implant threads (in its hypothetical exposure), a particulated bone graft (Creos Xenogain, Nobel Biocare AB) was used in the nasal cavity’s lateral portion and floor.

The surgical technique used for zygomatic implants’ placement considered the use of implants with a specific design: NobelZygoma 0° with a TiUnite surface (Nobel Biocare AB). The implant was placed following the procedures of the extramaxillary approach [19]. In this approach, a channel on the maxillary sinus was formed, allowing the drills direct access to the zygoma’s inferior edge. The Schneiderian membrane was reflected to prevent any damage and to avoid post-surgical sinus sequelae. The maxilla was not used for anchoring the implant, as it was anchored exclusively in the zygomatic bone. The implant’s prosthetic platform was shifted buccally to a more appropriate emergence position, typically between the first premolar and the first molar on the residual crest of the ridge, near its center [29].

Using a zygoma retractor (Modified Austin Tissue Retractor, Hu-Friedy, Germany) and with the surgeon’s thumb on the external surface of the zygoma’s upper edge to feel the preparation of the external cortical bone, the osteotomy began at a point as posterior as possible, keeping a safe 3 mm distance from the posterior vertical edge of the zygomatic bone. This position was used to reduce the cantilever and optimize biomechanics [30]. The orbit, the infra-orbital nerve, and the anatomy of the bone determined the drilling direction according to the pre-surgical planning with the 3D-printed model. A channel was created using a cylindrical diamond bur of 4 mm in diameter, with a tangential back-and-forth movement while increasing the depth. Following the channel creation, the round bur and the 2.9 mm twist drill (Nobel Biocare AB) were used in sequence. The correct implant length was selected using a depth indicator. According to the zygomatic bone density, the 3.5, 4.0, and 4.4 mm drills (Nobel Biocare AB) were used in sequence, followed by the implant insertion. After the implant placement and considering all the implants achieved at least 50 Ncm of primary stability, immediate function was performed with the connection of a temporary prosthesis on the same day as surgery. The 45-degree External Hex RP 6 mm and the 30-degree 5 mm multi-unit abutments (Nobel Biocare AB) were connected to the NobelZygoma 0° implants (Nobel Biocare), adjusting the mesial tilting of the implants and allowing the prosthetic screw access to be positioned on the occlusal aspect of the prosthetic teeth [31], using pre-made prostheses as a guide. For the transnasal implants, 30° and 17° 2.5 mm abutments (GM™ Mini Conical Abutment, Neodent) were chosen in order to compensate for the tilting of the pre-maxilla. The prosthetic screws were fastened using 15 Ncm torque.

To minimize the risk of vestibular soft tissue dehiscence with consequent exposure of the zygomatic implant body and threads, the implant head was covered with the buccal fat pad from Bichat’s ball. After suturing with non-resorbable sutures (Braun Silkam non-absorbable 4-0, Aesculap, Tuttlingen, Germany), open tray impression copings (Nobel Biocare AB) were placed and connected to provide an impression able to produce a model in which a high-density acrylic resin (Palaxpress Ultra, Heraeus Kulzer GmbH, Hanau, Germany) prosthesis, with temporary-coping multi-unit titanium (Nobel Biocare) inserted the same day and used during the healing period.

Once the patients reached the end of the healing period, the provisional prostheses were replaced by the final prostheses, designed and fabricated using CAD-CAM technology and comprising a titanium infrastructure (Nobel Biocare AB), artificial acrylic resin gingiva (Palaxpress Ultra, Heraeus Kulzer GmbH), and 12 acrylic-resin crowns (Mondial and Premium teeth, Heraeus Kulzer GmbH).

### 2.4. Post-Operative Interventions

Medication was prescribed for a two-week course: analgesics (Clonix 300 mg; Janssen-Cilag, High Wycombe, UK) were administered post-operatively for the first 3 days if needed, corticosteroid medication (Meticorten; Schering-Plough Farma Lda, Agualva-Cacém, Portugal) was administered daily in a regressive mode (30 mg at surgery and on the first 2 days post-operatively, 20 mg on days 3 and 4 post-operatively, 10 mg on day 5, 5 mg on days 6 and 7 post-operatively), anti-inflammatories (ibuprofen, 600 mg; Ratiopharm Lda, Carnaxide, Portugal) were given every 12 h on days 8 through 15 post-operatively, and antibiotics (amoxicillin, 875 mg, and clavulanic acid, 125 mg; Labesfal, Campo de Besteiros, Portugal) were given every 8 h daily for 7 days and every 12 h thereafter until day 15.

Post-operative follow-up visits were scheduled for 2 weeks after surgery and then at 2, 4, and 6 months. In all appointments, the acrylic-resin provisional prostheses were removed to perform the clinical assessments (implant mobility through manual assessment, suppuration by finger pressure, modified plaque index, modified bleeding index, and probing/mucosal seal efficacy evaluation), dental hygiene instructions, and prophylaxis (removal of bacterial plaque with a plastic tip ultrasonic scaler Instrument PI and Endo-chuck, EMS, Nyon, Switzerland; and polishing the abutments at the abutment–mucosa interface using a rubber cup with chlorohexidine gel attached to the contra-angle) [32]. The surgical, prosthetic, and maintenance procedures are illustrated in Figure 1, Figure 2 and Figure 3.

### 2.5. Outcome Measures

The patients were evaluated during the functional osseointegration period (from the day of implant surgery to the connection of the definitive prostheses, at between 6 and 10 months post-surgery): clinical evaluations were performed post-surgery at 10 days, 2 months, 4 months, and 6 months. The prostheses were always removed to perform clinical assessment and prophylaxis. The primary outcome measures were prosthetic and implant survival. The secondary outcome measures were biological and mechanical complications, plaque scores, scores, bleeding scores, and probing pocket depths/mucosal seal efficacy. Prosthetic survival was measured based on function, where the need to replace the prosthesis was considered a failure. Implant survival was also based on function, with the need to remove the implant (or presence of implant mobility) considered a failure. The modified plaque index [32] was used to evaluate the patient’s oral hygiene levels and recorded using an ordinal scale (0: absence of plaque; 1: plaque only visible after the passage of the periodontal probe around the peri-implant sulcus; 2: plaque visible by the naked eye; 3: abundance of soft matter). The bleeding scores were evaluated using the modified bleeding index [32], assessing the peri-implant health status, and recorded using an ordinal scale (0: absence of bleeding; 1: isolated bleeding spots visible; 2: bleeding forming a confluent line on the margin; 3: heavy or profuse bleeding). The complication parameters evaluated were biological (infection, fistula formation, or soft tissue inflammation) or mechanical (loosening or fracture of any prosthetic component). 

Descriptive statistics were used to assess the variables of interest, namely, the demographic variable age (average and standard deviation), the modified plaque index, and the modified bleeding index (mode). Frequencies were used to classify demographic variables and complication parameters. Implant survival was evaluated through life tables.

## 3. Results

### 3.1. Cases

Three female patients, with an average age of 62 years (standard deviation = 5.3 years), were treated. All patients presented comorbidities: patient 1, a 66-year-old patient with antidepressant and anxiolytic medication; patient 2, a 56-year-old female patient with musculoskeletal disease; and patient 3, a 64-year-old female patient with hypertension and a history of psychological complications. Table 1 characterizes the cases.

### 3.2. Prosthetic and Implant Survival

There were no prosthetic or implant failures throughout the functional osseointegration period, with us achieving a survival rate of 100%.

### 3.3. Complication Parameters, Plaque Scores, Bleeding Scores, and Probing Pocket Depths/Mucosal Seal Efficacy 

No biological complications occurred. Fracture of an acrylic crown in the provisional prosthesis occurred in patient 1 (crown #15) at 7 months of follow-up. The prosthesis was mended in the dental laboratory, and the occlusion was adjusted. No further mechanical complications occurred. The mode for mPLI was 1 (plaque only visible after passing the periodontal probe across the peri-implant sulcus) at 10 days and 2, 4, and 6 months, translating as mild plaque accumulation. The mode for mBI was 1 (isolated bleeding spots visible) at 10 days and 2, 4, and 6 months, translating as mild inflammation levels. No probing pocket depts over 4 mm were recorded in any implant during the follow-up.

## 4. Discussion

The present case series illustrated the digital planning, surgical/prosthetic protocols, and short-term follow-up of transnasal implants in conjunction with zygomatic implants for full-arch fixed prosthetic rehabilitation of atrophic maxillae using the *ad modum* all-on-4 concept (Nobel Biocare AB). 

Dental implant placement in the anterior maxillary alveolar ridge is limited by the nasal cavity, with various grafting techniques developed to overcome this situation. A maxillary inlay bone graft, interposition bone graft through Le Fort I osteotomy, and nasal floor augmentation by using autogenous bone or graft substitutes have been described as methods of reconstruction to allow the placement of standard dental implants in the atrophic pre-maxilla [33,34,35]. Preliminary reports about the nasal floor elevation suggest a two-step technique, using bone augmentation in a first stage and dental implant placement in a second stage [36,37]. Moreover, changes in grafting materials, implant technology, and surgical techniques and, recently, the incorporation of these new concepts into the nasal floor elevation technique have provided good survival and success rates for dental implants placed in the same surgical step [14,38]. Since 2003, immediate loading has been the main goal of implant treatment, especially in full-arch rehabilitation [39]; however, these techniques preclude the immediate function. So, the paranasal bone at the pyriform and the nasal crest have been used as key points of bone fixation in states of extreme atrophy by using angulated implants that are better able to access islands of cortical bone through parasinusal transfixation—the V-4 [40] approach. The zygoma-anchored implants have been successful in providing posterior prosthetic support. This technique, introduced and developed by Brånemark to avoid bone grafting, involves placing two to four conventional dental implants in the anterior maxilla in addition to the bilateral zygomatic implants [8,9,10,11,12,19]. Few studies have been performed reporting the survival rates of these anterior implants placed in combination with zygomatic implants: Vrielinck et al. [41], in a recent study with 1 to 20 years of follow-up, reported a cumulative survival rate of 67.7%, with no association between survival and the number of conventional implants, implying a viable treatment option even when the bone volume enables only two anterior implants to be inserted. Nevertheless, the authors registered an association between dental implants < 10 mm of length and decreased implant survival, suggesting treatment with quad zygoma if there is insufficient anterior bone to place implants >10 mm. 

The quad zygoma concept (all-on-4 double zygoma) came into clinical practice years after the single zygomatic implant (combined with conventional implants). The quad zygoma represents a highly demanding surgical technique that requires appropriate training, planning, and meticulous surgery. A recent long-term retrospective study on the quad zygoma concept [18] provided long-term evidence, registering a 97.8% implant survival rate, a 98.2% prosthesis success rate, and a mean score of 1.7 on oral health impact profile-14, translating as virtually “no impairment” on the patient’s quality of life self-assessment. Furthermore, a systematic review and meta-analysis by Aboul-Hosn Centenero et al. evaluated the survival rates of two zygomatic implants combined with regular implants versus four zygomatic implants, with no significant differences between the two treatment modalities in survival [42].

The insertion of extra-long transnasal implants could be considered a potential treatment alternative for patients with atrophic maxillae, provided sufficient bone volume for the apical locking of these implants in the frontal process of the maxilla [24]. The canine or fronto-maxillary pillar is an area with high cancellous and cortical bone density that acts as a protective frame for the nasal fossa. The lateral wall of the nasal cavity, where the nasal concha articulates with the internal surface of the maxilla, is a thickening that protrudes medially into the nasal cavity [43] but has still relatively thin dimensions [43,44]. 

In this technique, the extra-long implants were placed with high primary stability by tangential transfixation of the paranasal fossa and anchoring on that conchal crest [22,24]. Proper pre-surgical planning is mandatory, as in order to perform the technique, a minimum of 4 mm of residual bone on the crest is required, a nasal cavity not very pneumatized, and enough bone on the conchal crest (minimum 3 mm) [22,24]. The fact that this technique should not be used in patients presenting very wide nasal cavities or with a very distal lateral limit presents a limitation. Furthermore, the approach implies the surgeon having detailed anatomical knowledge of this region due to the proximity of the internal maxillary artery to the fronto-maxillary pillar and of the nasolacrimal duct in the turbinate.

The present technique included the use of a bone graft only to cover the exposed implant threads, avoiding direct contact between the implant and the nasal membrane and reducing the risk of implant exposure to the nasal cavity, rather than creating available bone for implant stability. This way, despite being a graft technique, some of the potential advantages linked to a graftless technique are allowed, such as morbidity reduction, a simplified surgical intervention, and a decreased operating time, all inherent strategies for ensuring immediate function [45,46,47,48].

Although the paranasal bone at the pyriform and nasal crest is still commonly present in stages of extreme atrophy [44], the use of this extra-long implant is limited to very extensive resorptions where the residual bone on the crest is less than 7 mm [26], hence the importance of the implant’s shape. Implant stabilization depends on the stabilization on the crest, and mainly on the anchorage of the apex (a thin area). The authors consider the use of narrow-diameter implants paramount, in order to reduce the amount of bone removed during the drilling procedure. 

The extramaxillary location of the zygomatic implant involves direct contact of the surface of the implant body with the mucoperiosteal flap. Some studies suggest an association between thinner mucosal phenotypes and increased peri-implant bone loss [49] and recession of the peri-implant mucosal margin [50]. However, other studies registered no association between the exposure of the zygomatic implant body and implant failure [19,21,51,52]. Nevertheless, as soft tissue dehiscence exposes the zygomatic implant body, this makes it more difficult for the patient to perform good self-care and increases the risk of additional complications such as sinus communication, mucositis, or cellulitis [19,51,52]. Protocols to prevent these potential complications could include the displacement of the buccal fat pad [53,54] over the zygomatic implant body, or the ZAGA Scar Graft [55]. In the present case series, the buccal fat pad was chosen given the harvesting simplicity of the approach with minor complications, easy manipulation, easy handling of the graft, anatomical proximity, and proven results [53,54,56]. Nevertheless, controlled studies are lacking that compare the two alternatives. 

The present short case series has provided an illustration of a potentially interesting alternative technique for full-arch atrophic maxillary rehabilitation. We have described a technique that is relatively recent and potentially simpler, and we hope that this starting point promotes more research, given that the technique warrants further investigation to be validated. Nevertheless, the anatomical inclusion criteria for the technique (requiring a minimum of 4 mm of residual bone on the crest, a nasal cavity not very pneumatized, and a minimum of 3 mm of bone on the conchal crest) limit its applicability to some patients. Furthermore, the limitations of the present investigation include it being performed in a single center and consisting of a short case series with short-term follow-up.

Future investigations should assess the long-term outcomes of rehabilitations using the present technique, using larger sample sizes and control groups.

## 5. Conclusions

This short case series of three patients has described the full-arch fixed prosthetic rehabilitation of atrophic maxillae using extra-long transnasal implants in combination with zygomatic implants in immediate function. This technique warrants more research to evaluate its clinical efficacy in the rehabilitation of patients with severely atrophic maxillae.

## Figures and Tables

**Figure 1 jcm-13-03348-f001:**
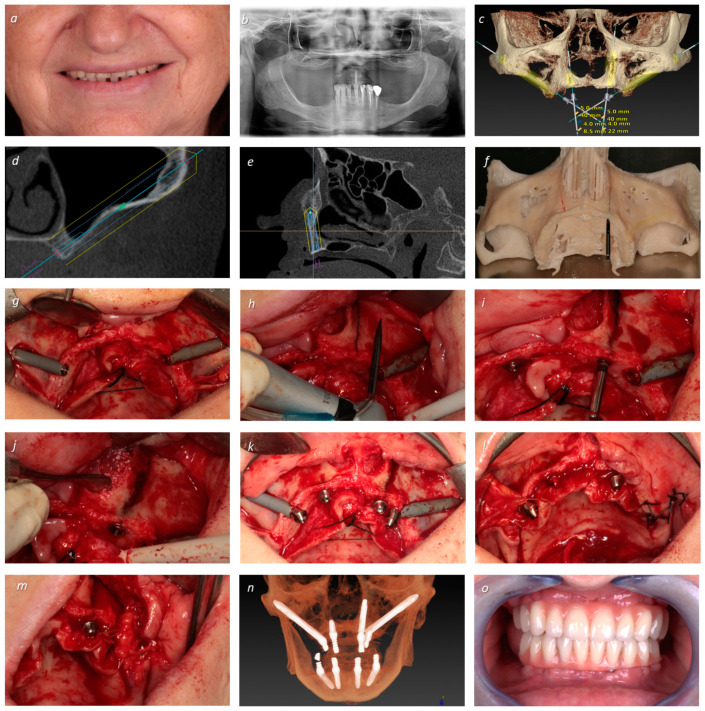
A 66-year-old female patient with antidepressant and anxiolytic medication complained of total missing teeth in upper jaw using a removable denture for 15 years. (**a**) Clinical examination revealed large resorption and loss of VOD; (**b**) orthopantomography revealing extensive bone loss on the maxilla with high pneumatization in both sites; (**c**) virtual implant planning, placing zygomatic implants (positions #15 and #25) combined with one conventional (position #12) and one transnasal implant (position #22); (**d**) virtual zygomatic implant planning; (**e**) virtual transnasal implant planning (NASA 3 classification) [24] (note the nasolacrimal canal); (**f**) stereolithographic model planning; (**g**) intra-oral peri-operative view with insertion of two NobelZygoma 0° with 40 mm of length (Nobel Biocare AB) in positions #15 and #25; (**h**) drilling pathway for the transnasal implant preparation; (**i**) insertion of the transnasal implant in position #22 (Helix GM^®^ Long 3.75 × 22.5 mm, Neodent, Curitiba, Brazil), where less than 4 mm of bone was available; (**j**) performing a bone graft (Creos, Nobel Biocare AB) along the internal extra-osseous portion of the transnasal implant; (**k**) intra-oral peri-operative view after connection of multi-unit abutments (Nobel Biocare AB): 30° and 5 mm of height (zygomatic implant in position #15), 45° and 6 mm of height (zygomatic implant in position #25), 17° and 3 mm of height (standard implant in position #12), 30° and 2.5 mm of height (transnasal implant in position #22); Bichat ball used as graft over the extra-osseous portion of the zygomatic implants: (**l**) implant in position #15; (**m**) implant in position #25; (**n**) post-operative CBCT displaying the bimaxillary rehabilitations performed in the immediate function; (**o**) frontal post-operative view of the rehabilitations with connection of the provisional prosthesis on the same day of surgery, achieving immediate function.

**Figure 2 jcm-13-03348-f002:**
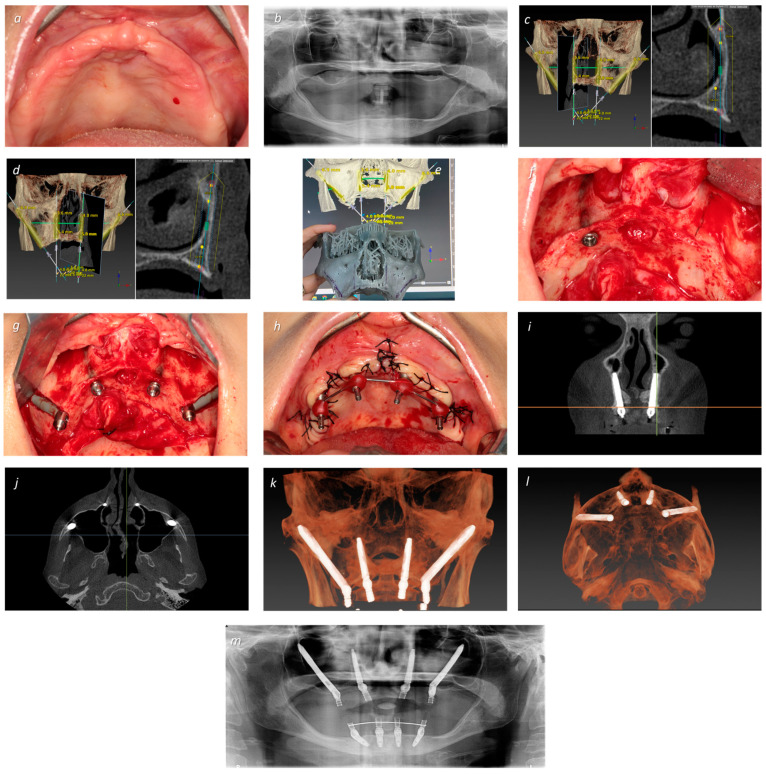
A 56-year-old female patient with musculoskeletal disease undergoing treatment with physiotherapy and analgesic medication, complaining of painful removable prosthesis and absolute loss of retention. (**a**) Intra-oral pre-operative occlusal photograph of the maxilla exhibiting significant resorption; (**b**) orthopantomography revealing significant resorption and high sinus pneumatization disabling the insertion of short implants in the anterior maxilla; (**c**) transnasal right implant planning in position #12 (NASA 3 classification) [24] (note also the planned position of the right zygomatic implant in position #15); (**d**) transnasal left implant planning in position #22 (NASA 3 classification) [24] (note also the planned position of the left zygomatic implant in position #25); (**e**) surgical simulation on the stereolithographic model (prototype of the middle third of the face with extended maxilla); (**f**) intra-oral peri-operative view of the right transnasal implant placed, and the left transnasal implant pathway marked with a pencil; (**g**) intra-oral peri-operative view with insertion of two NobelZygoma 0° (Nobel Biocare AB) in positions #15 (40 mm of length) and #25 (40 mm of length), and two transnasal implants in positions #12 and #22 (Helix GM^®^ Long 3.75 × 22.5 mm, Neodent, Curitiba, Brazil) with all multi-unit abutments connected; (**h**) intra-oral peri-operative photograph after suturing and before impressions, with the impression copings attached to the abutments; (**i**) post-operative CBCT slice exhibiting the final frontal transnasal implant’s position; (**j**) post-operative CBCT slice exhibiting the axial view of the final transnasal apex implant’s position; (**k**) post-operative CBCT frontal view of the final implant rehabilitation; (**l**) post-operative CBCT axial view of the final implant rehabilitation; (**m**) post-operative CBCT of the final bimaxillary rehabilitation achieving immediate function.

**Figure 3 jcm-13-03348-f003:**
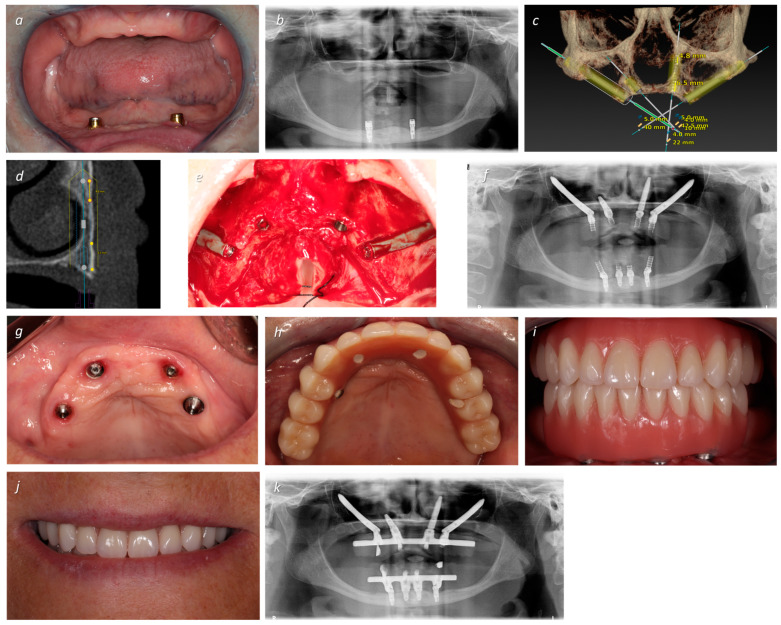
A 64-year-old female patient with hypertension and a history of psychological complications, with lack of retention of both removable prostheses. (**a**) Frontal pre-operative intra-oral photograph (note the large resorption and loss of VOD); (**b**) pre-operative orthopantomography revealing severe bone loss and inadequate bone in the maxillary sinus and anterior region; (**c**) virtual implant planning, placing zygomatic implants (positions #15 and #25) combined with one conventional mesially-tilted implant (position #12), aiming for increased bone contact surface, and one transnasal implant (position #22); (**d**) transnasal implant planning with the DTX software (V3.6.4.2; NASA 3 classification) [24], where CBCT slices reveal that this implant is anchored in 6.5 mm of the crest and 4.8 mm of the nasal concha; (**e**) intra-oral peri-operative view with insertion of two NobelZygoma 0° with 40 mm of length (Nobel Biocare AB) in positions #15 and #25, one NobelSpeedy Groovy RP 10 mm (Nobel Biocare AB) in position #12, and one Helix GM^®^ Long 3.75 × 22.5 mm (Neodent, Curitiba, Brazil) in position #22; (**f**) post-operative orthopantomography displaying the bimaxillary rehabilitations performed in immediate function; (**g**) occlusal post-operative intra-oral photograph displaying the healing at 4 months of follow-up; (**h**) occlusal intra-oral photograph displaying the definitive prosthesis (a titanium infrastructure with acrylic resin artificial gingiva and crowns); (**i**) frontal intra-oral photograph displaying the definitive prosthesis; (**j**) patient smiling with the definitive prosthesis; (**k**) final orthopantomography with the bimaxillary rehabilitations.

**Table 1 jcm-13-03348-t001:** Case characterization in relation to implant position of emergence, implant characteristics, opposing dentition, and presence in follow-up appointments.

Patient	Location of Implant Emergence				Age	Sex	NASA	OD	MP	IS
	Right		Left							
	Second Premolar	Lateral Incisor	Lateral Incisor	Second Premolar						
1	EM 0° 5 × 40	S 4 × 8.5	T 3.75 × 22.5	EM 0° 5 × 40	66	F	3	I	NG	+50
2	EM 0° 5 × 40	T 3.75 × 22.5	T 3.75 × 22.5	EM 0° 5 × 40	56	F	3	I	NG	50
3	EM 0° 5 × 40	S 4 × 10	T 3.75 × 22.5	EM 0° 5 × 40	64	F	3	I	NG	+50

Type of implant: EM—extramaxillary zygomatic implant; T—transnasal implant; S—standard implant; diameter × length (mm); NASA—Nasal Anatomic Systematic Approach: 0: crestal approach from the foramen; 1: crestal approach from central incisor; 2: crestal approach from lateral incisor; 3: crestal approach from canine; 4: crestal approach from canine with a screw thread exposed [23]. OD—opposing dentition: I—implant-supported prosthesis; N—natural teeth; R—removable prosthesis; 4—(2 + 3); MP—method of placement: guided (G), non-guided (NG); IS—initial stability in Ncm.

## Data Availability

Data will be provided by the authors upon reasonable request.

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
