# Peer review of "All-on-4 Hybrid with Extra-Long Transnasal Implants: Descriptions of the Technique and Short-Term Outcomes in Three Cases"

_jcm, 2024, doi:10.3390/jcm13113348_

Round 1
Reviewer 1 Report
Comments and Suggestions for Authors
Abstract:
“No implants nor prostheses failed achieving a 100% survival rate” The double negative in this sentence is confusing. Please rephrase to make it easier to understand.
Materials and methods:
“The maxilla was not used for anchoring the implant, as it was anchored exclusively in the zygomatic bone.” Please explain how this was achieved and verified.
“To minimize the risk of vestibular soft tissue dehiscence ……the implant head was covered with buccal fat pad from Bichat´s ball” What was the rationale of using fat bad and not other more proven graft types?
Please provide the type of suture material used.
Discussion:
“This way, it can be considered as a graftless technique” I disagree with this statement. A graftless technique implies no use of any grafting materials regardless of the aim of the grafting procedure. Please describe the procedure in another, more accurate term or omit it altogether.
Minor issues:
Please mention the full abbreviation of WHO in text.
“Once the patients overcame the healing period…” overcame is not an appropriate word choice. Please change.
“Preliminary reports (34, 35 e 36) about the nasal floor elevation….” Please correct the in-text citations.
Author Response
Abstract:
- “No implants nor prostheses failed achieving a 100% survival rate” The double negative in this sentence is confusing. Please rephrase to make it easier to understand.
Response: The authors thank the Reviewer’s indication. The phrase was amended as requested.
Changes: Abstract section, lines 23 and 24
Materials and methods:
- “The maxilla was not used for anchoring the implant, as it was anchored exclusively in the zygomatic bone.” Please explain how this was achieved and verified.
Response: The authors thank the Reviewer’s query. The zygomatic implant was inserted using the extramaxillary surgical approach, where the apex is anchored on the zygomatic bone and the coronal portion is accommodated in the maxilla in a channel and covered only by soft tissue. The authors refer to reference number 23 and further demonstrate it in figures 1 g), i), k); 2 g) and 3 e).
Changes: None
- “To minimize the risk of vestibular soft tissue dehiscence ……the implant head was covered with buccal fat pad from Bichat´s ball” What was the rationale of using fat bad and not other more proven graft types?
Response: The authors thank the Reviewer’s query. The Bichat’s ball was used as an alternative to the connective tissue graft given the simplicity of the approach with minor complications, easy manipulation and handling of the graft, anatomical proximity, and also proven results. The authors do not question the merits of connective soft tissue graft in resolving soft tissue dehiscences, but would like to highlight that this procedure was not performed to resolve a soft tissue dehiscence, but to prevent soft tissue dehiscence. In this sense, the use of the Bichat’s ball in zygomatic implants soft tissue dehiscence prevention (de Morais 2012; Guennal and Guiol 2018) predates the use of connective tissue which was only recently introduced (Aparício and Antonio 2020). Furthermore, no systematic reviews nor meta-analysis exist on the comparison between both modalities. Given the interesting topic raised by the Reviewer, the discussion was further developed on the two alternatives to the one already present in lines 406-407.
References used in this reply:
- de Moraes EJ. The buccal fat pad flap: an option to prevent and treat complications regarding complex zygomatic implant surgery. Preliminary report. Int J Oral Maxillofac Implants. 2012 Jul-Aug;27(4):905-10.
- Guennal P, Guiol J. Use of buccal fat pads to prevent vestibular gingival recession of zygomatic implants. J Stomatol Oral Maxillofac Surg. 2018 Apr;119(2):161-163. doi: 10.1016/j.jormas.2017.10.017. Epub 2017 Nov 3.
- Aparicio C, Antonio S. Zygoma Anatomy-Guided Approach "Scarf Graft" for Prevention of Soft Tissue Dehiscence Around Zygomatic Implants: Technical Note. Int J Oral Maxillofac Implants. 2020 Mar/Apr;35(2):e21-e26. doi: 10.11607/jomi.8065.
Changes: Discussion section, lines 410-414
- Please provide the type of suture material used.
Response: The authors thank the Reviewer’s query. The suture material type was provided as requested.
Changes: Materials and Methods section, lines 189 and 190.
Discussion:
- “This way, it can be considered as a graftless technique” I disagree with this statement. A graftless technique implies no use of any grafting materials regardless of the aim of the grafting procedure. Please describe the procedure in another, more accurate term or omit it altogether.
Response: The authors thank the Reviewer’s comment and agree. For the sake of clarity, the sentence was changed as it is a graft technique that allowed for some of the potential advantages linked to graftless approaches.
Changes: Discussion section, lines 390-392.
Minor issues:
- Please mention the full abbreviation of WHO in text.
Response: The authors thank the Reviewer’s indication. The full abbreviation of WHO was indicated in the text as requested.
Changes: Introduction section, lines 34.
- “Once the patients overcame the healing period…” overcame is not an appropriate word choice. Please change.
Response: The authors thank the Reviewer’s indication. The word was replaced as requested.
Changes: Materials and Methods section, line 195
- “Preliminary reports (34, 35 e 36) about the nasal floor elevation….” Please correct the in-text citations.
Response: The authors thank the Reviewer’s correction. The in-text citations were deleted.
Changes: Discussion section, line 339
Reviewer 2 Report
Comments and Suggestions for Authors
Thank you for the opportunity to review a paper on All-on-4 Hybrid with extra-long transnasal implants. This surgical technique gives effective oral rehabilitation of patients with severe bone resorption of the maxilla. However, because of the advanced surgical procedures, reported success rates and prognoses vary, and evidence needs to continue to accumulate. Therefore, this paper will shed light on the future of oral implant therapy. The following points need to be improved to make it more reader-friendly.
Authors should remove the period at the end of the title. The content category of this paper seems to be case report, not article.
The transnasal implant technique has been in use for many years. The author should introduce more about the novelty of this method in the introduction in order to present this paper. What is new information for the reader (e.g., improved surgical procedures? prognosis?) should be indicated.
In the Materials and Methods section, surgical methods are described in detail and preferred. However, there is little information on initial fixation at the time of implant placement. What was the value of the ISQ? The fastening torque of the prosthetic screw should also be noted.
In the results section, individual cases are presented with many pictures, which is preferred. However, a new table should be added to allow easy comparison of the three cases. For example, age, gender, implants used, method of placement, prognosis, etc.
Discussion Section
There are few cases to accumulate evidence regarding this technique, and the course of the disease is short. It is unclear what the authors want to appeal with this reported content. At the end of the section, a generalized discussion should be given from the findings obtained. Limitation should also be considered and described.
Author Response
Thank you for the opportunity to review a paper on All-on-4 Hybrid with extra-long transnasal implants. This surgical technique gives effective oral rehabilitation of patients with severe bone resorption of the maxilla. However, because of the advanced surgical procedures, reported success rates and prognoses vary, and evidence needs to continue to accumulate. Therefore, this paper will shed light on the future of oral implant therapy. The following points need to be improved to make it more reader-friendly.
- Authors should remove the period at the end of the title. The content category of this paper seems to be case report, not article.
Response: The authors thank the Reviewer’s indication. The period was deleted as requested. The authors assume this article as a case series.
Changes: Title section, line 3.
- The transnasal implant technique has been in use for many years. The author should introduce more about the novelty of this method in the introduction in order to present this paper. What is new information for the reader (e.g., improved surgical procedures? prognosis?) should be indicated.
Response: The authors thank the Reviewer’s query. The authors would like to point that the current technique is transnasal and not transinus. The Transnasal technique is a technique that was first documented in 2019 by one of the co-authors of this paper (Vanderlim). After a through Pubmed search, it was possible to find two citations close to 2019:
- One citation in 2021 by Almeida et al. that refers to it as “a new treatment option” and refers to the cited publication by Vanderlim in 2019 (that we added now);
- One citation in 2023 by Sahin, configuring a case report.
The manuscript deals with the surgical procedure and digital planning achieved for the rehabilitation. A phrase was included in the first sentence of the Discussion section for clarity.
References used in this reply:
- Almeida PHT, Cacciacane SH, Arcazas Junior A. Extra-long transnasal implants as alternative for Quad Zygoma: Case report. Ann Med Surg (Lond). 2021 Jul 27;68:102635. doi: 10.1016/j.amsu.2021.102635.
- Åžahin O. Treatment of Severely Atrophic Maxilla by Using Zygomatic, Pterygoid, and Transnasal Implants. J Craniofac Surg. 2023 Nov 20. doi: 10.1097/SCS.0000000000009896.
Changes: Discussion section, lines 327-328.
- In the Materials and Methods section, surgical methods are described in detail and preferred. However, there is little information on initial fixation at the time of implant placement. What was the value of the ISQ? The fastening torque of the prosthetic screw should also be noted.
Response: The authors thank the Reviewer’s queries. All implants achieved an initial stability of at least 50 Ncm: one patient all 4 implants achieved and initial stability of 50 Ncm, all the other 8 implants in the two patients achieved an initial stability of more than 50 Ncm. The fastening torque of the prosthetic screws was 15 Ncm as per manufacturer indications.
Changes: Materials and Methods section, lines 178,185,186
- In the results section, individual cases are presented with many pictures, which is preferred. However, a new table should be added to allow easy comparison of the three cases. For example, age, gender, implants used, method of placement, prognosis, etc.
Response: The authors thank the Reviewer’s indications. The table was inserted as requested.
Changes: Results section, Table 1, line 311
Discussion Section
- There are few cases to accumulate evidence regarding this technique, and the course of the disease is short. It is unclear what the authors want to appeal with this reported content. At the end of the section, a generalized discussion should be given from the findings obtained. Limitation should also be considered and described.
Response: The authors thank the Reviewer’s query. The discussion was inserted as requested.
Changes: Discussion section, lines 430-440.
Round 2
Reviewer 2 Report
Comments and Suggestions for Authors
I appreciate the chances you made.